# Role of Ischemia/Reperfusion and Oxidative Stress in Shock State

**DOI:** 10.3390/cells14110808

**Published:** 2025-05-30

**Authors:** Yarielis Ivette Vázquez-Galán, Sandra Guzmán-Silahua, Walter Ángel Trujillo-Rangel, Simón Quetzalcoatl Rodríguez-Lara

**Affiliations:** 1School of Medicine International Program, Universidad Autónoma de Guadalajara, Av. Patria 1201, Zapopan 45129, Jalisco, Mexico; yarielisi.vazquez@edu.uag.mx (Y.I.V.-G.); sandra.silahua@edu.uag.mx (S.G.-S.); 2Unidad de Investigación Epidemiológica y en Servicios de Salud, Centro Médico Nacional de Occidente Órgano de Operación Administrativa Desconcentrada Jalisco, Instituto Mexicano del Seguro Social, Guadalajara 44329, Jalisco, Mexico; 3Departamento de Aparatos y Sistemas II, Decanato de Ciencias de la Salud, Universidad Autónoma de Guadalajara, Av. Patria 1201, Lomas del Valle, Zapopan 45129, Jalisco, Mexico; walter.trujillo@edu.uag.mx; 4Departamento de Ciencias Biomédicas, Centro Universitario de Tonalá, Universidad de Guadalajara, Tonalá 45425, Jalisco, Mexico

**Keywords:** shock, ischemia/reperfusion injury, oxidative stress, immune dysregulation, HIF pathway, inflammatory response, multi-organ dysfunction

## Abstract

Shock is a life-threatening condition characterized by inadequate tissue perfusion leading to systemic hypoxia and metabolic failure. Ischemia/reperfusion (I/R) injury exacerbates shock progression through oxidative stress and immune dysregulation, contributing to multi-organ dysfunction. This narrative review synthesizes current evidence on the interplay between I/R injury, oxidative stress, and immune modulation in shock states. We analyze the classification of shock, its progression, and the molecular pathways involved in ischemic adaptation, inflammatory responses, and oxidative injury. Shock pathophysiology is driven by systemic ischemia, triggering adaptive responses such as hypoxia-inducible factor (HIF) signaling and metabolic reprogramming. However, prolonged hypoxia leads to mitochondrial dysfunction, increased reactive oxygen species (ROS) and reactive nitrogen species (RNS) production, and immune activation. The transition from systemic inflammatory response syndrome (SIRS) to compensatory anti-inflammatory response syndrome (CARS) contributes to immune imbalance, further aggravating tissue damage. Dysregulated immune checkpoint pathways, including CTLA-4 and PD-1, fail to suppress excessive inflammation, exacerbating oxidative injury and immune exhaustion. The intricate relationship between oxidative stress, ischemia/reperfusion injury, and immune dysregulation in shock states highlights potential therapeutic targets. Strategies aimed at modulating redox homeostasis, controlling immune responses, and mitigating I/R damage may improve patient outcomes. Future research should focus on novel interventions that restore immune balance while preventing excessive oxidative injury.

## 1. Introduction

The shock state (a life-threatening condition caused by inadequate blood flow) is a critical condition produced by an insufficient supply of oxygen and nutrients to the tissues in relation to the tissue metabolic demand [1,2,3]. The pathological state progresses with the deterioration of the function of vital organs such as the brain, heart, kidneys, lungs, liver, and gastrointestinal tract. For instance, reduced perfusion in the kidneys may lead to acute kidney injury, while hypoxia in the brain can result in cognitive impairment or loss of consciousness [4,5,6,7,8,9]. The secondary effect is mediated by circulatory failure that will produce inadequate or inappropriate tissue perfusion distribution resulting in systemic cellular hypoxia. Under these critical conditions, the pathophysiological characteristics are driven by inadequate oxygen and metabolic substrate supply, alongside increasing demands for these requirements. This imbalance can result in cellular injury and eventual organ dysfunction [10,11,12,13,14]. In parallel, the inability to eliminate metabolites and wastes resulting from energy expenditure, hypoxic adaptation, cellular injury, and cell death can exacerbate the condition, ultimately causing permanent damage [15,16,17,18].

Understanding any shock state requires recognizing the concept of shock as “inadequate organ and peripheral tissue perfusion” [1,2,3,18,19,20,21,22,23,24,25,26]. When analyzing ischemia/reperfusion (I/R) lesions, it is important to note that ischemia is defined as “the abrupt blockage of the blood supply that causes an imbalance in the oxygen supply and metabolic nutrients essential for cell survival”. This leads to hypoxia, metabolic disruption, and impaired energy production [10,11,13,14,26,27,28]. The shock state and the first component of I/R lesions share the same outcome: systemic cellular hypoxia and metabolic failure. Although the mechanisms behind their initiation may differ, both phenomena result in similar detrimental effects across the entire system [10,11,13,14,26,27,28].

## 2. Classification and Categorization of Shock State

To understand the process of shock, it is necessary to classify the different types, as they differ in pathological mechanisms and therapeutic approaches. These differences are crucial because they guide the selection of appropriate treatments. For instance, hypovolemic shock caused by acute blood loss requires fluid resuscitation and blood transfusion, whereas septic shock requires antibiotics to target the underlying infection and vasopressors to restore vascular tone [29,30,31]. Shock is currently classified into six types (Table 1): hypovolemic, distributive, cardiogenic, obstructive, and mixed. Each type has distinct causes and requires specialized management [1,3,19,20,21,29,30].

Hypovolemic shock is a condition where the system presents inadequate organ perfusion caused by loss of intravascular volume [1,2,3,19,20,32,33,34]. It can be subclassified into four main categories depending on the mechanics of the lesion (hemorrhagic, traumatic/hemorrhagic, pure hypovolemic, or traumatic/hypovolemic) (Table 1). This type of shock primarily arises from a critical reduction in circulating volume, which impairs venous return and diminishes cardiac preload, ultimately leading to a drop in cardiac output. In the hemorrhagic subtype, blood loss occurs rapidly, either externally or internally, and can result from vascular rupture, trauma, or surgical complications. In traumatic hemorrhagic cases, the combination of bleeding and tissue damage triggers an inflammatory response and further fluid sequestration into the interstitial space. Pure hypovolemic shock, on the other hand, is caused by severe fluid loss without hemorrhage—often through persistent vomiting, diarrhea, or heat-related dehydration—leading to hypovolemia with minimal tissue injury. Traumatic hypovolemic shock involves similar fluid depletion but is compounded by soft tissue injury or burns, which promote capillary leakage and third-spacing. Regardless of the etiology, all hypovolemic shock subtypes converge on the common pathophysiologic outcome of inadequate end-organ perfusion and generalized tissue ischemia, requiring prompt volume resuscitation to restore hemodynamic stability [1,2,3,19,20,32,33,34].

Distributive shock is the critical redistribution of the absolute intravascular volume and, depending on its causes, can be subclassified into four major types: septic (infections), anaphylactic (immune response), neurogenic (acute neurological trauma), and endocrine shock (acute adrenal insufficiency) (Table 1) [1,2,3,19,20,30,31,35,36,37]. Septic shock is distinct due to its infectious etiology, leading to widespread inflammation and vasodilation triggered by microbial toxins. In contrast, anaphylactic shock results from a severe allergic reaction, where histamine release causes rapid vascular permeability and hypotension.

Neurogenic shock involves loss of sympathetic tone, often after spinal cord injury, leading to unopposed parasympathetic activity and bradycardia. Endocrine shock, such as in acute adrenal insufficiency, results from hormonal deficits causing vascular instability. These differences underline the importance of precise identification for targeted management [1,2,3,19,20,30,31,35,36,37].

Cardiogenic shock is the critical reduction of the heart’s pumping capacity; the most common causes are myocardial failure (acute myocardial infarction), cardiac conduction system failure (brady and tachyarrhythmias), and heart valve dysfunctions (acute insufficiency and decompensated stenosis) (Table 1) [1,2,3,19,20,38,39,40,41].

Obstructive shock occurs due to obstruction in critical vascular or cardiac structures. It can arise from extracardiac conditions such as aortic dissection, which impede blood flow, or mechanical obstructions affecting the heart, like tumors or hemopericardium. Additionally, issues with afterload or preload may impair venous return or increase resistance, as seen in pneumothorax, hemothorax, pneumopericardium, or hemopericardium. Pulmonary causes, such as pulmonary embolism, can also hinder blood flow in the lungs. Each of these mechanisms results in inadequate cardiac output and systemic perfusion, necessitating targeted interventions [1,2,3,19,20,41,42].

Dissociative or cytotoxic shock (Diss/Cyto) arises from the inability of tissues to utilize oxygen effectively, despite adequate oxygen delivery and normal or near-normal hemodynamic parameters (Table 1). This condition is typically caused by toxic or metabolic insults that impair cellular respiration at the mitochondrial level. Examples include poisoning with cyanide or carbon monoxide, which disrupt oxidative phosphorylation by inhibiting cytochrome c oxidase or displacing oxygen from hemoglobin, respectively. In such scenarios, oxygen may be present in sufficient quantity in the bloodstream, yet it remains biologically unavailable to cells. As a result, aerobic metabolism fails, leading to a shift toward anaerobic glycolysis, lactate accumulation, and widespread cellular dysfunction. This form of shock is distinct in that it represents a failure of oxygen utilization, rather than oxygen delivery, and requires rapid identification and specific antidotal therapy to prevent irreversible organ damage [1,2,3,19,20,41,42].

Besides the classification of shock and subclass, there is a categorization of shock severity that involves the ability of the body to compensate. Non-progressive shock, also known as compensated shock, is characterized by the activation of allostatic mechanisms, such as increased heart rate and vasoconstriction, to maintain perfusion to vital organs. Progressive shock occurs when these compensatory mechanisms fail, leading to worsening tissue hypoxia, metabolic acidosis, and organ dysfunction (Figure 1) [1,2,3,19,20,21,22,24,25,43,44,45,46,47,48,49]. Finally, irreversible shock results in multi-organ dysfunction syndrome (MODS), where tissue damage becomes severe and unresponsive to therapeutic interventions [50,51,52,53]. In the compensated state, the mechanisms of allostasis (the body’s process of maintaining stability) temporarily adapt to the pathological changes induced by the shock state. During this period, unaffected organs and physiological systems strive to maintain perfusion to vital organs. However, if these adaptations fail, the system transitions into a non-compensatory state, leading to further deterioration and progression of the shock state [1,2,3,19,20,21,22,24,25,43,44,45,46,47,48,49].

## 3. Progression of Shock State

Independently of the type of shock, the progression will have a stereotyped development where, depending on the ability of the tissues to tolerate the ischemia, the degree of the initial injury, and the delay of the initial treatment to containment/eliminate the aggression, the non-injured tissue will start to present lesion or develop permanent damage [1,2,3,19,20,21,22,24,25,43,44,45,46,47,48,49]. In the clinical scenario, there is the presence of multiple syndromes that follow the initial shock phase, which follows the intent of the body to adapt to the aggression (Figure 1). However, if the shock progresses, it will produce deleterious effects on the prognosis of the patient [54,55,56].

In the cellular scenario, several events perpetuate the progression of the shock, decreasing the ability of the cells to tolerate the aggression. The micro-verse (cellular neighboring) and macro-verse (organ systems intercommunication) will orchestrate together the death of the cells and the body system failure.

## 4. The Micro-Verse

In the progression of cellular function decay, the primary aggressor is systemic ischemia, which affects all cell populations [26,57]. Certain groups—such as endothelial cells, mesangial cells, and inflammatory cells—activate adaptive responses (adrenergic, hormonal, metabolic, and hypoxia-inducible pathways) that increase their tolerance to ischemic injury; these are considered cells with high ischemic tolerance [58,59,60,61,62,63]. Tissues such as skeletal muscle, skin, bone, and hepatocytes generally exhibit greater resistance to the metabolic changes associated with early injury and systemic ischemia [64,65,66,67,68,69,70]. In contrast, cells with inherently lower ischemic tolerance—including epithelial cells, endothelial cells in sensitive vascular beds, cardiomyocytes, neurons, previously injured cells, or those affected by chronic comorbidities such as diabetes, hypertension, cancer, or toxic exposures—are more prone to entering pathological states such as apoptosis, autophagy, or necrosis [71,72,73,74,75,76].

The dynamic interaction between these cellular territories—those with high and low ischemic tolerance—shapes the early pathogenesis of shock, even before the onset of reperfusion injury [26,57,58,59,60,61,62,63,64,65,66,67,68,69,70,71,72,73,74,75,76]. The duration of systemic ischemia following the initial insult, along with incomplete or inadequate resolution of the ischemic state, remains a critical determinant of prognosis. Prolonged hypoperfusion and intermittent I/R episodes promote irreversible cellular damage and propagate systemic dysfunction (Figure 2) [77,78,79,80,81,82,83].

While this review does not aim to explore the full range of cellular responses to ischemia and reperfusion—which would warrant independent investigation—it emphasizes the relevance of the micro-verse: the localized, cell-specific network of tolerance and vulnerability that precedes and shapes systemic outcomes. Understanding this landscape is key to elucidating how cellular-level dynamics drive the progression and clinical manifestations of the shock state (SIRS domain, CARS domain, MARS domain, and CHAOS) (Figure 2) and identifying targets for intervention that could improve long-term outcomes.

### 4.1. Adaptative Micro-Verse System During Shock Progression

Modern healthcare systems employ highly standardized training protocols for the early recognition and treatment of shock, which has significantly reduced the duration of systemic ischemia compared to previous decades [77,78,79,80,81,82,83]. While these strategies have improved immediate survival outcomes, they often focus solely on resolving the initial insult—specifically, hypoperfusion and oxygen debt—without addressing the deeper cellular consequences. However, shock is increasingly understood as an I/R phenomenon involving three key cellular segments of injury: (1) activation of membrane and metabolic processes aimed at tolerating intracellular changes due to energetic substrate deprivation (e.g., cytosolic cation influx, oxidative stress, mitochondrial dysfunction); (2) intercellular signaling and interaction among different cell types (neighboring effect) that propagate damage (e.g., endothelial injury, no-reflow phenomenon, transcriptional reprogramming); and (3) systemic I/R lesions characterized by immune activation, apoptosis, autophagy, and necrosis. Unlike isolated arterial occlusions, which typically localize damage to a specific organ, systemic shock states affect multiple organs and systems simultaneously [27,84,85,86].

Specific groups of cells—those primarily affected during the first hit—undergo membrane destabilization and generate large quantities of reactive oxygen species (ROS), thereby amplifying surrounding damage through neighboring effects. These include endothelial cells, epithelial cells, cardiomyocytes, renal tubular cells, neurons, and resident immune cells. Depending on their inherent ischemic tolerance, these cell populations encounter the I/R lesion at different time points in an asynchronous and tissue-specific manner, leading to distinct clinical manifestations (e.g., arrhythmias, seizures, immune dysregulation, fever, renal failure, capillary dysfunction, pulmonary gas exchange abnormalities, hepatic dysfunction) [11,84,85,86,87,88,89]. Cells capable of initiating adaptive responses engage molecular pathways that enhance their resistance to hypoxia—these are considered cells with high ischemic tolerance. One of the most critical regulatory networks is the hypoxia-inducible factor (HIF) pathway, which orchestrates the transcriptional adaptation to hypoxic stress [90,91,92]. Unfortunately, current clinical paradigms rarely integrate the dynamics of these cellular adaptations into therapeutic strategies. By overlooking this microenvironmental resilience, shock management risks become incomplete, targeting only systemic metrics without addressing the cellular groundwork that may delay or accelerate injury progression.

### 4.2. The Role of HIF During Shock and I/R

The HIF pathway consists of transcription factors that regulate cellular adaptation, with three major members: HIF-1 (HIF-1α & HIF-1β), HIF-2α, and HIF-3α. The functional dynamics of HIF-1 are complex as it interacts with other HIF family members in a tissue-specific manner. While HIF-1α and HIF-1β are ubiquitously expressed in all cells, HIF-2α is primarily found in epithelial cells of the lungs and endothelial cells in the carotid body. In contrast, HIF-3α is predominantly expressed in Purkinje cells of the cerebellum and corneal cells [92,93,94,95]. HIF-1α and HIF-2α share approximately 48% sequence homology and can dimerize with HIF-1β to interact with hypoxia response elements (HREs) in DNA, thereby modulating gene transcription. While HIF-1α and HIF-2α enhance gene expression, HIF-3α serves as an inhibitor of HRE-mediated gene transcription [92,93,94,95].

Under hypoxic conditions, HIF-1α is stabilized and activated through phosphorylation by MAPK signaling. It subsequently forms a complex with CBP/p300, which translocates into the nucleus. There, it dimerizes with HIF-1β, forming the HIF-1α(CBP/p300)/HIF-1β complex, which binds to hypoxia-response elements (HREs) in DNA. This binding initiates the transcription of genes involved in erythropoiesis, iron metabolism, angiogenesis, glucose metabolism, cell proliferation, survival, and apoptosis. The extent and specificity of gene expression depend on the cell type involved (e.g., endothelial, myocardial, epithelial, immune, neuronal, skeletal muscle, pluripotent) and the duration of hypoxic/ischemic activation (ranging from seconds to hours) [92,93,94,95,96]. Among the most critical functions of HIF signaling is the induction of angiogenesis, primarily mediated by HIF-1α and HIF-2α isoforms. HIF-1α strongly promotes the transcription of key pro-angiogenic genes, including vascular endothelial growth factor (VEGF), stromal-derived factor 1 (SDF-1), and angiopoietin-1 (Ang-1) [97,98]. These factors unequivocally facilitate neovascularization and promote the restoration of blood flow in ischemic tissues. Meanwhile, HIF-2α, predominantly expressed in endothelial cells, regulates genes such as Tie2 and angiopoietin-2 (Ang-2), which are essential for vascular maturation and stabilization [99]. The complementary and distinct roles of HIF-1α and HIF-2α enable a precisely orchestrated angiogenic response that adapts to both acute and chronic hypoxic conditions [97,98,99].

In shock states, cell populations with high ischemic tolerance—such as skeletal muscle cells, hepatocytes, and fibroblasts—initially respond by shifting their metabolism toward oxygen-independent glycolysis. This adaptive response involves the upregulation of glucose transporters (GLUT-1 and GLUT-3) and the enhancement of glycolytic enzyme activity to sustain ATP and pyruvate production. Concurrently, the HIF-1α/CBP/p300/HIF-1β complex promotes the expression of key metabolic genes, including lactate dehydrogenase A (LDHA), monocarboxylate transporter 4 (MCT4), pyruvate dehydrogenase kinase 1 (PDK1), COX4-2, and the mitochondrial protease LONP1. This metabolic reprogramming facilitates lactate accumulation (which is subsequently exported via MCT4), inhibits the conversion of pyruvate to acetyl-CoA (through PDK1-mediated inhibition of PDH), attenuates oxidative phosphorylation, and ultimately reduces excessive mitochondrial production of ROS and reactive nitrogen species (RNS) by enhancing electron transport efficiency via COX4-2 modulation and LONP1-mediated COX4 degradation [100,101,102].

Beyond its metabolic effects, HIF activation also has profound implications in angiogenesis, particularly during ischemia/reperfusion injury and in various forms of shock. In such settings, the restoration of microvascular blood flow is vital for tissue survival, and HIF signaling plays a central role in promoting revascularization and maintaining endothelial homeostasis [97,98]. However, it is essential to recognize that persistent or dysregulated HIF activity may have deleterious effects, including increased capillary permeability, excessive neovascularization, and the formation of dysfunctional vasculature, all of which can exacerbate tissue injury and compromise organ function [97].

Beyond its metabolic effects, HIF-1α signaling promotes cellular proliferation and survival by inducing the expression of insulin-like growth factor-2 (IGF-2) and transforming growth factor-alpha (TGF-α), alongside activating the MAPK, PI3K, and AMPK pathways. These responses exhibit both local (autocrine) and systemic (endocrine) effects, a phenomenon termed the “neighboring effect”. These mechanisms collectively help keep cells within a “point of safe return”, the threshold before mitochondrial damage becomes irreversible [11,13,14,27].

However, in cell populations with low ischemic tolerance—such as neurons, cardiomyocytes, renal tubular epithelial cells, and endothelial cells in vulnerable vascular territories—or when the shock state persists, the excessive activity of the HIF pathway may lead to maladaptive outcomes. These include the induction of apoptosis, overactivation of the immune system, and upregulation of adrenergic receptors. Clinically, this maladaptation is associated with arrhythmias, blood pressure instability, and progressive organ dysfunction [103,104]. Overactive HIF signaling enhances the expression of pro-apoptotic genes, including caspase-3, Fas/Fas-ligand, Bcl-2/adenovirus EIB19, BNip3, and NIX, and further amplifies p53 and p21 signaling pathways. This cascade promotes the expression of apoptosis mediators such as Bax, NOXA, PUMA, and PERP, ultimately leading to widespread cellular death and late-stage tissue injury [100,101,102].

In septic shock, HIF-1α can be activated even in the absence of severe hypoxia, primarily via inflammatory mediators such as lipopolysaccharides (LPSs), tumor necrosis factor-alpha (TNF-α), and interleukin-1β (IL-1β). This activation exerts a dual effect: while it enhances glycolytic flux in immune and endothelial cells as an adaptive survival mechanism, it simultaneously exacerbates mitochondrial dysfunction, inflammation, and vascular permeability, contributing to tissue damage and poor outcomes [105].

Conversely, in hemorrhagic shock, where hypoxia results directly from impaired tissue perfusion, HIF-1α activation initiates compensatory mechanisms that help preserve cellular viability. These include the upregulation of VEGF and antioxidant enzymes, which are crucial for maintaining endothelial integrity and promoting vascular repair during the reperfusion phase [98,105].

A key clinical marker of deteriorating HIF-mediated adaptation is oxygen debt, which correlates with elevated blood lactate levels. In the clinical setting, sustained lactate elevation is strongly associated with shock progression and poor prognosis [2,39,54,55,106,107,108]. This reflects ongoing anaerobic metabolism, increased oxygen demand by hypoxic tissues, and insufficient systemic perfusion to meet metabolic needs. Lactate monitoring thus provides valuable prognostic information and serves as a guide for therapeutic decision-making aimed at mitigating I/R injury [2,39,54,55,106,107,108].

Taken together, these findings highlight the dual and time-dependent nature of HIF signaling during shock and ischemia/reperfusion. While early HIF activation supports cell survival, metabolic adaptation, and vascular regeneration, its prolonged or dysregulated expression contributes to irreversible tissue injury and systemic failure. Thus, HIF represents a pivotal regulatory hub—both a defender and a potential driver of damage—depending on the cellular context, type of shock, and duration of hypoxia. Understanding this balance is essential not only for unraveling the molecular basis of shock progression but also for identifying therapeutic windows in which modulating HIF activity may improve outcomes and prevent long-term complications.

## 5. The Macro-Verse

As the shock state progresses and the ability of metabolically adaptive cell populations to tolerate systemic ischemia begins to deteriorate, organ systems become interdependent, coordinating compensatory responses in an effort to sustain global physiological function [81,82,83,84,109]. Several systemic regulatory axes are activated, most notably the adrenergic, endocrine, and immune systems. Initially, these responses act synergistically to preserve perfusion to vital organs—such as the brain, heart, lungs, liver, and kidneys—through mechanisms including the release of adrenaline and noradrenaline, activation of adrenergic receptors, stimulation of the renin–angiotensin–aldosterone system (RAAS) to modulate vascular tone and fluid balance, and immune-mediated repair signaling. However, as cellular damage accumulates and the duration of ischemia/reperfusion extends, these compensatory systems begin to falter. The immune response, in particular, becomes dysregulated, shifting from a reparative role to a pathological driver of inflammation, tissue injury, and systemic dysfunction [14,58,59,61,110]. This marks the transition from cellular adaptation to multi-organ compromise, laying the foundation for the immunological syndromes that will be explored in the following sections.

### 5.1. Ischemia Phase and Immune System

During the ischemic phase of shock progression, as previously described, multiple cellular stress-response pathways are activated in an attempt to enhance ischemic tolerance. Despite these adaptive efforts, once the I/R injury is initiated, a cascade of immune clinical syndromes emerges, each representing the body’s attempt to respond and adapt to systemic aggression [111,112,113,114]. However, if the extent of ischemic damage surpasses the system’s compensatory capacity, the immune response transitions from a repair-and-sustain role to one of degradation and destruction, ultimately promoting apoptosis and inflammatory damage in vulnerable organs [52,53,54,55,57,113,114,115,116].

As shock advances, the immune system undergoes a staged transformation (Figure 3), typically progressing through four major phases: (i) Systemic Inflammatory Response Syndrome (SIRS), characterized by widespread immune activation and massive release of pro-inflammatory cytokines; (ii) Compensatory Anti-inflammatory Response Syndrome (CARS), a regulatory phase aimed at suppressing excessive inflammation and restoring immune equilibrium; (iii) Mixed Antagonistic Response Syndrome (MARS), where simultaneous pro- and anti-inflammatory signals lead to immune imbalance and functional dysregulation; and (iv) CHAOS—an acronym for Cardiovascular Compromise, Loss of Homeostasis, Apoptosis, Organ Dysfunction, and Immune Suppression—which represents a state of systemic collapse and immune exhaustion [53,57,113,114,117,118,119,120,121]. Each of these immune syndromes reflects a dynamic and context-dependent adaptation of the immune system to persistent ischemia, oxidative stress, and metabolic instability, ultimately shaping clinical outcomes and guiding potential therapeutic strategies [53,57,113,114,117,118,119,120,121].

The activation of the immune system begins with the first hit (the initial insult), where the pathological mechanisms of each shock type and subclass play a role in triggering immune responses. Affected cells upregulate the expression of pattern recognition receptors (PRRs), including Pathogen-Associated Molecular Patterns (PAMPs), which are released when the mucosae (such as the skin, eyes, genital tract, or gastrointestinal tract) are disrupted, exposing constitutional microbiota and surrounding pathogens to the systemic circulation [122,123,124,125,126,127,128,129,130,131]. Damage-Associated Molecular Patterns (DAMPs) originate from locally traumatized tissue, while PAMPs are characteristic of septic shock, where microbial invasion drives immune activation [122,123,124,125,126,127,128,129,130,131,132]. These signals collectively stimulate the production of key inflammatory cytokines, including IL-1, IL-6, IL-17, TNF-alpha, and IL-10, while also activating NF-κB signaling [122,123,124,125,126,127,128,129,130,131,132].

Initially, immune activity aims to contain damage and facilitate tissue repair through the SIRS mechanism. However, if the injury is extensive or the shock state progresses, excessive immune activation can lead to a dysregulated response. This results in the transition from SIRS to CARS, a counter-regulatory mechanism designed to suppress excessive inflammation and restore homeostasis (Table 2) [122,123,124,125,126,127,128,129,130,131,132].

As the progression or extension of damage in shock continues, the interaction between the SIRS and CARS responses induces the state of MARS, representing a dynamic balance between these opposing immune responses (Figure 2). Clinically, this phase corresponds to the compensated state of shock, where all organs and tissues remain in a precarious equilibrium that can only persist for a limited period before reaching a critical threshold of failure. If this threshold is exceeded, the system may shift toward either of two pathological overlays: (i) SIRS dominance over MARS, which exacerbates tissue destruction and coagulation activity mediated by the immune system (Figure 3), or (ii) CARS dominance over SIRS, increasing susceptibility to infections and delaying tissue repair. In both scenarios, the system ultimately deteriorates into CHAOS, leading to MODS and systemic failure (Figure 3) [53,57,113,114,117,118,119,120,121].

The immune components play a crucial role in regulating various clinical syndromes, particularly as shock progresses (Figure 3). Understanding the functional dynamics of these cellular components is essential for modulating immune responses and mitigating pathological outcomes.

#### 5.1.1. IL-1 Signaling Pathway: Activation and Inhibition

The interleukin-1 receptor (IL-1R) family consists of ten members classified into four subgroups: ligand-binding receptors (IL-1R1, IL-1R2, IL-1R4, IL-1R5, and IL-1R6), accessory proteins (IL-1R3 and IL-1R7), negative regulatory receptors (IL-1R2, IL-1R8, and IL-1R8BP), and members with unknown functions (IL-1R9 and IL-1R10) [133,134,135,136,137,138,139]. These receptors play a critical role in immune regulation by interacting with Toll-like receptors (TLRs) through the Toll/interleukin-1 receptor (TIR) domain, which is essential for recruiting and differentiating immune cells. Furthermore, the TIR domain shares homology with MyD88, a key adaptor molecule involved in immune signaling pathways (Figure 4) [133,134,135,136,137,138,139,140,141].

IL-1R1 activation occurs when it forms a complex with accessory proteins IL-1R3 and IL-1R7, allowing the binding of ligands such as IL-1α, IL-1β, or IL-38. The downstream effects of IL-1R activation depend on the target cell type and include the induction of inflammatory cytokines, amplifying immune responses [133,134,135,136,137,138,139,140,141]; the generation of ROS and RNS, contributing to cellular damage [135,142,143,144]; increased prostaglandin synthesis, promoting inflammatory mediator production [136,145,146]; proteolytic enzyme activation, which facilitates extracellular matrix degradation and tissue remodeling [146,147,148,149,150,151,152,153,154]; and immune system modulation, enhancing adaptive immune responses through T cell expansion and Th17 differentiation [155,156,157,158].

At the intracellular level, IL-1R signaling begins when a ligand, such as IL-1β, binds to IL-1R1, forming a receptor–ligand complex with IL-1R3. This interaction recruits MyD88 via the TIR domain, leading to the activation of two major signaling cascades: the nuclear factor kappa B (NF-κB) pathway, which drives the transcription of proinflammatory cytokines [133,134,135,136,137,138,139,140,141], and the mitogen-activated protein kinase (MAPK) pathway, which regulates inflammatory mediator production and cellular stress responses [158,159,160,161]. While these pathways are essential for responding to infections and tissue injury, their uncontrolled activation can result in chronic inflammation, tissue damage, and autoimmune disorders [158,159,160,161,162].

Overactivation of IL-1R signaling contributes to oxidative stress by promoting excessive ROS and RNS production. Mitochondria are the primary sources of ROS, while NADPH oxidase plays a critical role in generating superoxide radicals [163,164,165,166]. This oxidative imbalance leads to mitochondrial dysfunction, impairing energy production and promoting apoptosis [163,164,165,166]; endoplasmic reticulum (ER) stress, disrupting protein folding and triggering the unfolded protein response [141,167,168]; lysosomal damage, resulting in the release of hydrolytic enzymes that contribute to cell death; and DNA damage, leading to genotoxic stress, mutations, and cellular senescence [169,170,171].

Certain cell types are particularly susceptible to excessive IL-1 signaling and oxidative stress. Immune cells such as macrophages, neutrophils, and dendritic cells may become hyperactivated, leading to a state of uncontrolled inflammation [172,173,174]. Endothelial cells experience increased vascular permeability and dysfunction, contributing to systemic inflammation [175,176,177,178,179]. Neurons are particularly vulnerable to chronic inflammation and oxidative damage, which have been implicated in neurodegenerative diseases [180,181,182]. Cardiomyocytes also suffer from oxidative stress-induced injury, exacerbating heart failure progression [183,184,185].

In clinical conditions such as shock and ischemia/reperfusion injury (IRI), IL-1R activation plays a pivotal role in tissue damage. During shock, excessive cytokine production and oxidative stress drive systemic inflammation and MODS [186,187]. In ischemia, oxygen deprivation triggers metabolic distress, while reperfusion further exacerbates injury through a surge in ROS production and inflammatory mediators [188,189,190].

To counteract excessive IL-1 signaling, the immune system employs regulatory mechanisms such as soluble and decoy receptors that sequester IL-1 ligands. Among them, sIL-1R2 and sIL-1R3 function as competitive inhibitors by binding IL-1 ligands without initiating downstream signaling [191,192]. However, in prolonged shock or ischemic conditions, the excessive expression of decoy receptors can lead to immunosuppression, increasing susceptibility to secondary infections and impairing recovery [177,193,194,195]. This phenomenon is particularly evident in post-cardiac arrest syndrome and septic shock, where dysregulated IL-1 signaling has been associated with poor clinical outcomes [196,197,198,199].

Maintaining a balanced IL-1 signaling response is crucial for immune homeostasis and preventing excessive inflammation. Future research should focus on developing therapeutic interventions targeting this pathway to mitigate tissue damage in inflammatory and ischemic conditions.

#### 5.1.2. IL-6 and TNF-α Pathways in Shock Progression

The IL-6 family consists of 10 ligands and nine receptors, with signaling mediated by the gp130 receptor, which is ubiquitously expressed in all cells [200,201,202]. The pathway involves Janus kinase (JAK), the signal transducer and activator of transcription factor 3 (STAT3), and JAK-SHP-2-mitogen-activated protein kinase (MAPK). IL-6 interacts with IL-6Rα (membrane-bound and soluble forms) and recruits gp130 to initiate intracellular signaling. This signaling cascade modulates inflammation, endothelial activation, hepatic acute-phase protein production, and immune cell differentiation [203,204,205,206,207,208,209].

IL-6 signaling can contribute to oxidative stress by activating intracellular pathways that lead to the increased production of ROS. The activation of the STAT3 and MAPK pathways has been shown to enhance mitochondrial ROS production, leading to oxidative damage and further perpetuation of inflammatory responses [210,211,212,213,214]. Additionally, IL-6 can induce the expression of NADPH oxidase (NOX) enzymes, which catalyze the production of ROS, exacerbating oxidative stress and promoting endothelial dysfunction [215,216,217,218].

Excessive IL-6 signaling leads to chronic inflammation, tissue damage, and immune exhaustion, whereas inadequate signaling results in impaired immune responses and susceptibility to infections [219,220,221].

The TNF family includes at least 18 ligands and 29 receptors, mediating inflammation, apoptosis, and immune system regulation. TNF-α interacts with TNFR1 and TNFR2, leading to differential downstream signaling. TNFR1 activation predominantly results in proinflammatory and apoptotic pathways, while TNFR2 signaling is associated with immune modulation and tissue repair [222,223,224].

In early shock response, TNF-α/TNFR1 and TNF-α/TNFR2 drive macrophage activation, monocyte recruitment, and cytokine amplification [225,226,227]. While essential in initial host defense, excessive TNF-α signaling contributes to endothelial dysfunction, tissue damage, and systemic inflammation characteristic of septic shock, ischemia/reperfusion injury, and multiple organ dysfunction syndrome (MODS) [228,229,230,231,232,233]. Additionally, TNF-α induces oxidative stress by stimulating mitochondrial dysfunction and increasing ROS production, exacerbating cellular damage [234,235,236,237,238,239].

The therapeutic inhibition of TNF pathways is widely explored in inflammatory diseases such as rheumatoid arthritis and inflammatory bowel disease, yet excessive downregulation in critically ill patients can impair immune defense and increase susceptibility to secondary infections [240,241,242,243].

#### 5.1.3. Integration of IL-1, IL-6, and TNF-α in Inflammatory Waves

Under synergistic action, IL-1, IL-6, and TNF-α drive the initial inflammatory wave aimed at damage containment and repair (Figure 4). This wave is characterized by (i) pro-inflammatory actions, including rapid cytokine production, immune cell recruitment, and the activation of tissue repair mechanisms, and (ii) self-modulatory mechanisms, such as the simultaneous release of regulatory molecules, including decoy receptors, to limit overactivation.

The regulation of these pathways is critical to maintaining immune balance. Excessive activation, as seen in chronic inflammation or severe injuries, can lead to tissue damage and autoimmune conditions. Conversely, the overexpression of regulatory mechanisms (e.g., decoy receptors) may result in immunosuppression, especially in prolonged shock states. This duality underscores the importance of (i) the therapeutic targeting of pathways (e.g., IL-1 receptor antagonists, JAK inhibitors) and (ii) monitoring cytokine levels to predict progression from pro-inflammatory to regulatory phases.

The balance between IL-6-, TNF-α-, and IL-1-driven inflammation determines shock progression. Uncontrolled proinflammatory signaling leads to tissue damage, while excessive counter-regulation via soluble cytokine receptors or anti-inflammatory mediators may cause immune paralysis and secondary infections [244,245,246,247]. Understanding these dynamics is crucial for developing targeted immunomodulatory therapies in critical care medicine.

#### 5.1.4. CTLA-4 and PD-1: Immune Checkpoint Pathways

CTLA-4 (cytotoxic T-lymphocyte antigen 4) and PD-1 (programmed cell death protein-1) are key regulators of immune response suppression and inflammation control and can be overactivated or inhibited by several mechanisms (Table 3) [248,249,250,251,252,253,254,255,256]. They function as immune checkpoints that prevent excessive immune activation and protect against autoimmunity, but, in anomalous conditions, they can have a negative contribution regarding the survival of a patient in a shock progression state.

CTLA-4 competes with CD28 for B7 ligands (CD80/CD86), suppressing T-cell activation by limiting costimulatory signaling. It recruits phosphatases such as SHP-2 and PP2A, which dephosphorylate key signaling proteins like CD3 and ZAP70, preventing full T-cell activation (Figure 5). The dominance of its activation is upregulated in the presence of an excessive immune response [257,258].

PD-1 is expressed on activated T-cells, B cells, and monocytes. When engaged with its ligands, PD-L1 or PD-L2, PD-1 recruits SHP-1 and SHP-2 to dephosphorylate ZAP-70, blocking the downstream activation of the PI3K-Akt and Ras-MEK-ERK pathways. This results in the inhibition of T-cell proliferation and cytokine release [259,260,261,262].

T cell exhaustion and immune checkpoint engagement typically occur during the compensatory (CARS) and mixed (MARS) phases of shock, rather than immediately after the first insult, reflecting a delayed but critical stage of immune regulation. Future research should explore the role of non-conventional immune checkpoints (e.g., Siglec, LAIR-1, CD200R, KIRs), particularly in regulating innate and stromal cell responses during shock and I/R injury, which may uncover novel therapeutic avenues beyond the canonical PD-1/CTLA-4 axis.

### 5.2. Integration of Inflammatory/Anti-Inflammatory Signaling

The signaling pathways of IL-1, IL-6, and TNF-α play a central role in activating the immune system and regulating inflammatory responses. However, immune checkpoint mechanisms such as CTLA-4 and PD-1 modulate these signals to prevent excessive and potentially harmful immune reactions [252,253,255,263,264,265,266].

CTLA-4 exerts its regulatory effect by dampening IL-1-driven inflammatory responses. It achieves this by inhibiting early T-cell activation, a key process in the amplification of IL-1-mediated inflammation. By blocking this initial activation, CTLA-4 limits the production of pro-inflammatory mediators, thereby reducing tissue damage associated with exaggerated immune responses [252,253,255,263,264,265,266].

On the other hand, the PD-1 pathway plays a crucial role in modulating IL-6 signaling. PD-1 activation inhibits IL-6-induced STAT3 activation, leading to a reduction in inflammatory cytokine production and a lower propensity for cytokine storms. This mechanism is essential in preventing uncontrolled systemic inflammation, which can result in MODS and severe tissue damage [252,253,255,263,264,265,266].

Furthermore, both CTLA-4 and PD-1 work together to counteract TNF-α signaling. TNF-α is a key cytokine in inflammation and immune activation, but its excessive activity can lead to tissue damage and immune exhaustion. The regulation of TNF-α by these immune checkpoint pathways helps maintain a balance between an effective immune response against pathogens and the prevention of excessive inflammation or autoimmunity [252,253,255,263,264,265,266]. Stromal cells, including fibroblasts, pericytes, and mesenchymal stromal cells, contribute to tissue remodeling and immune modulation during I/R injury by releasing regulatory cytokines and ECM components and influencing immune checkpoint pathways through paracrine signals.

Thus, immune checkpoints CTLA-4 and PD-1 play essential roles in maintaining immune homeostasis, preventing excessive inflammatory responses that could compromise tissue integrity and organ function. Although the identity of specific antigens in shock remains unclear, immune checkpoint activation may occur through antigen-independent chronic stimulation via DAMPs and inflammatory mediators, leading to T cell dysfunction and immune suppression.

## 6. Oxidative Stress and Shock States

In conditions like ischemia/reperfusion injury and shock, immune checkpoint dysregulation exacerbates oxidative stress and tissue damage [69,234,267,268,269]. Oxidative stress plays a critical role throughout the entire pathophysiology of shock, from the initial insult to the progression of the shock state and the establishment of ischemia/reperfusion (I/R) injury. Redox signaling is deeply involved in this process, and even after the resolution of shock, oxidative stress remains active, influencing either recovery or progression into CHAOS [270,271,272]. Under physiological conditions, oxidative stress is essential for proper cellular function, regulating various signaling pathways [273,274]. However, in the shock state, oxidative stress production becomes overwhelming due to the extent of cell damage from the initial insult, systemic ischemia, immune system activation, and the massive release of ROS and RNS during reperfusion [69,234,267,268,269,275]. The oxidative burst not only impacts injured cells but also affects non-damaged and repairing cells, amplifying systemic dysfunction.

Although oxidative stress is necessary for cellular function at a low level, its functional threshold is very narrow. Any perturbation can enhance its activity and induce an imbalance, although the body possesses enzymatic and non-enzymatic mechanisms to regulate oxidative homeostasis [273,274]. In a clinical setting, patients typically maintain an oxidative balance, fluctuating within normal limits. However, certain populations, including individuals with diabetes, hypertension, dyslipidemia, and cancer, often live under chronic oxidative stress conditions, making them particularly vulnerable to oxidative stress-induced injury [276,277,278,279,280,281].

In shock states, we propose five major factors that could determine the extent of oxidative stress-mediated injury: the degree of cellular damage during the first insult, the level of immune system activation, the impact of I/R injury, the presence of pre-existing pathological conditions, and the basal oxidative stress levels of the patient. If all five components are severe or persist over prolonged periods, oxidative stress leads to extensive cellular dysfunction, affecting survival and worsening the prognosis of shock patients. While some tissues exhibit higher ischemic tolerance, immune and endothelial cells are major ROS and RNS producers [271,282,283,284,285,286]. Furthermore, highly metabolically active cells contribute significantly to the oxidative burden.

The presence of oxidative stress in shock states serves a dual role in regulating inflammation. On the one hand, it promotes the activation of proinflammatory pathways. ROS and RNS activate NF-κB by degrading its inhibitor, IκB, leading to the upregulation of inflammatory cytokines such as IL-1, IL-6, and TNF-α [276,287,288,289,290,291,292]. These cytokines further amplify oxidative stress by inducing NADPH oxidase activation in macrophages and neutrophils. Additionally, ROS stimulate the NLRP3 inflammasome, triggering caspase-1 activation and the maturation of IL-1β and IL-18 [276,287,288,289,290,291,292]. This process creates a self-sustaining loop in which oxidative stress perpetuates inflammation. Moreover, excessive ROS and RNS production induce immunogenic cell death mechanisms, such as necroptosis and pyroptosis, causing the release of damage-associated molecular patterns (DAMPs), including HMGB1, ATP, and mitochondrial DNA. These DAMPs activate toll-like receptors (TLRs) and pattern recognition receptors (PRRs), further amplifying the inflammatory response [124,293,294,295,296,297].

On the other hand, oxidative stress also suppresses mechanisms that regulate inflammation, thereby preventing the resolution of immune activation. One of the primary mechanisms of this suppression is the disruption of PD-1 and CTLA-4 regulatory functions [298,299,300,301]. ROS and RNS impair the expression and function of these immune checkpoint proteins, reducing their ability to suppress immune activation. PD-1 normally inhibits T-cell activation by recruiting SHP-1 and SHP-2, but oxidative stress inactivates these phosphatases, allowing unchecked inflammation to persist. Additionally, oxidative stress inhibits IL-10 and TGF-β production, two critical cytokines required for inflammation resolution. The suppression of IL-10 expression in M2 macrophages and dendritic cells contributes to chronic inflammation, further exacerbating tissue damage [302,303,304,305,306].

Oxidative stress also alters the balance between Tregs and Th17 cells, which play an essential role in immune homeostasis. Under normal conditions, Tregs function to suppress excessive immune activation. However, ROS and RNS favor Th17 differentiation by activating STAT3 and RORγT, leading to increased inflammation [307,308,309,310]. Simultaneously, oxidative stress destabilizes Foxp3, a key transcription factor required for Treg differentiation, further tipping the balance toward proinflammatory responses. The resulting immune dysregulation can contribute to persistent inflammation, tissue destruction, and increased susceptibility to secondary infections [311,312,313].

Ischemia/reperfusion injury introduces an additional layer of complexity to the oxidative stress response. During the ischemic phase, hypoxia induces HIF-1α, which upregulates inflammatory genes and promotes anaerobic metabolism [102,314,315,316,317]. This metabolic shift leads to mitochondrial damage, releasing cytochrome C and mitochondrial DNA, which activate PRRs and amplify inflammation. Upon reperfusion, a massive oxidative burst occurs as oxygen re-enters the ischemic tissues, leading to mitochondrial ROS overproduction. This sudden influx of ROS triggers NF-κB activation and NLRP3 inflammasome stimulation, perpetuating a self-sustaining inflammatory cycle [318,319,320].

The interplay between oxidative stress, dysregulated inflammation, and ischemia/reperfusion injury has significant clinical implications (Table 4). Excessive ROS and RNS production contribute to MODS, endothelial damage, and coagulopathy. The uncontrolled activation of NF-κB and inflammasomes leads to cytokine storm development in septic shock. The loss of PD-1 and CTLA-4 function results in persistent immune activation and tissue destruction. The disruption of the Treg/Th17 balance fosters chronic inflammation and increases the risk of opportunistic infections. Furthermore, reperfusion-induced oxidative overload worsens ischemia/reperfusion injury and increases patient mortality [321,322,323].

## 7. Conclusions

Given the profound role of oxidative stress in the pathophysiology of shock, therapeutic strategies aimed at reducing ROS and RNS production hold promise for improving clinical outcomes. Potential interventions include NF-κB inhibition using antioxidants such as N-acetylcysteine and flavonoids, the modulation of the NLRP3 inflammasome using pharmacologic inhibitors like MCC950, the reactivation of PD-1 and CTLA-4 pathways to control excessive immune activation, targeting mitochondrial ROS production with agents such as mitochondria-targeted antioxidant agents to prevent reperfusion injury, and restoring the Treg/Th17 balance through therapies that modulate STAT3 and Foxp3 expression.

Understanding the interplay between oxidative stress, inflammation, and ischemia/reperfusion injury provides insight into novel treatment approaches for managing shock states. By targeting oxidative stress-mediated mechanisms, clinicians may be able to mitigate excessive inflammation, reduce tissue damage, and improve overall patient survival. Further research into these therapeutic avenues may lead to the development of effective interventions that balance immune control while preserving essential inflammatory responses necessary for tissue repair and recovery.

## Figures and Tables

**Figure 1 cells-14-00808-f001:**
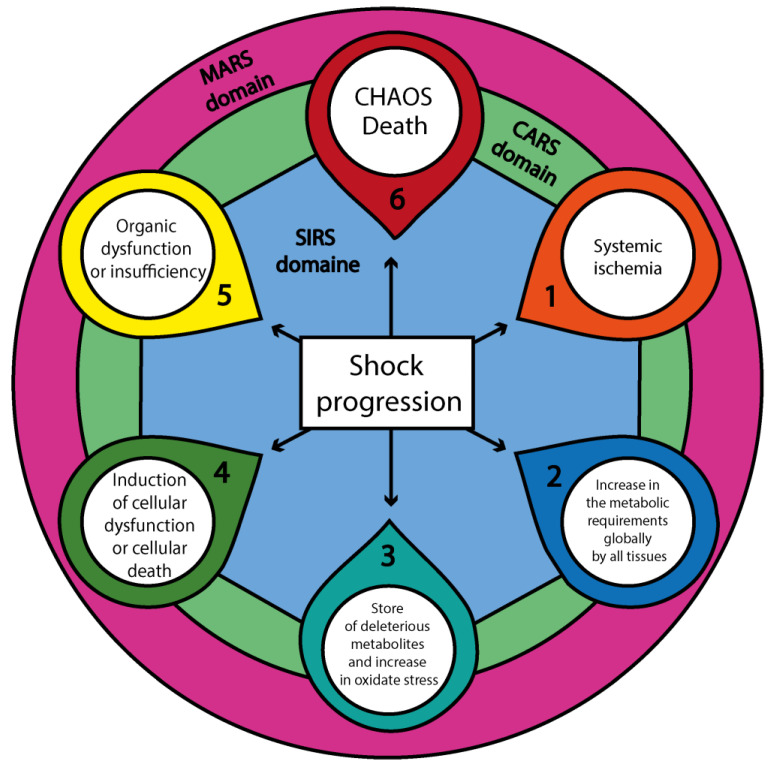
The progression of shock and its systemic immunological domains. This diagram represents the sequential pathophysiologic progression of shock, beginning with systemic ischemia (1), followed by a global increase in cellular metabolic demands (2), the accumulation of deleterious metabolites and oxidative stress (3), the onset of cellular dysfunction or death (4), and organ dysfunction or failure (5), culminating in CHAOS (6): Cardiovascular Compromise—Loss of Homeostasis—Apoptosis—Organ Dysfunction (MODS, SOF, MOF)—Immune Suppression. These six components radiate outward from a central white core labeled “Shock progression”, which anchors the diagram and symbolizes the primary pathological axis. This core is surrounded concentrically by three immunological domains: a blue ring representing SIRS (Systemic Inflammatory Response Syndrome), a green ring representing CARS (Compensatory Anti-inflammatory Response Syndrome), and an outer purple ring denoting MARS (Mixed Antagonistic Response Syndrome). These domains illustrate the immunometabolic milieu—the dynamic environment in which immune and metabolic responses interact and evolve—while also supporting localized intercellular communication through neighboring effects among cellular populations. Together, these processes modulate or amplify each step of shock progression, reinforcing the multiscale nature of systemic failure. Abbreviations: CHAOS—Cardiovascular Compromise, Homeostatic Loss, Apoptosis, Organ Dysfunction, Immune Suppression; MODS—Multiorgan Dysfunction Syndrome; SOF—Single Organ Failure; MOF—Multiple Organ Failure; SIRS—Systemic Inflammatory Response Syndrome; CARS—Compensatory Anti-inflammatory Response Syndrome; MARS—Mixed Antagonistic Response Syndrome.

**Figure 2 cells-14-00808-f002:**
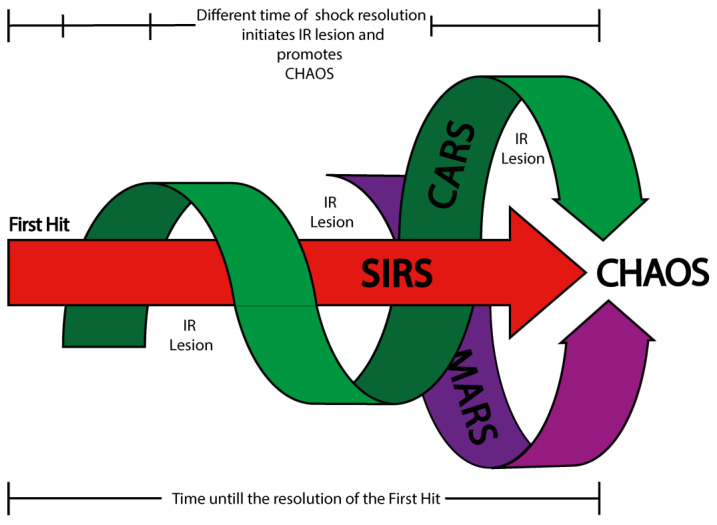
Shock, ischemia/reperfusion (I/R) injury, and immune syndrome domains. This diagram illustrates the temporal interplay between shock progression, ischemia/reperfusion (I/R) injury, and immune system syndromes. Following the first hit (the initial insult that establishes the shock state), a stereotyped immune response is initiated, beginning with the activation of the Systemic Inflammatory Response Syndrome (SIRS). Slightly delayed, but interwoven with SIRS, the Compensatory Anti-inflammatory Response Syndrome (CARS) emerges, followed by the Mixed Antagonistic Response Syndrome (MARS) in later phases. These three immune domains are depicted as interlinked arrows converging toward CHAOS (while progression toward CHAOS reflects systemic deterioration and immune exhaustion, not a distinct second hit), an acronym representing Cardiovascular Compromise, Loss of Homeostasis, Apoptosis, Organ Dysfunction (MODS, SOF, MOF), and Immune Suppression. Above the segmented black bar is symbolized the fluctuating nature of systemic ischemia, with multiple gaps in the arrows indicating intermittent periods of I/R injury. These recurrent ischemia/reperfusion events exacerbate systemic damage and promote immune dysregulation, contributing to the development of CHAOS. Below, a secondary time bar reflects the duration required to resolve the first hit; the longer this period, the greater the exposure to cumulative I/R episodes. If shock resolution is delayed or partial, temporally uncoupled I/R injuries may occur, intensifying the inflammatory burden and accelerating the immune syndromes. This model emphasizes that systemic damage is not solely driven by immune activation but also by the timing and severity of reperfusion dynamics.

**Figure 3 cells-14-00808-f003:**
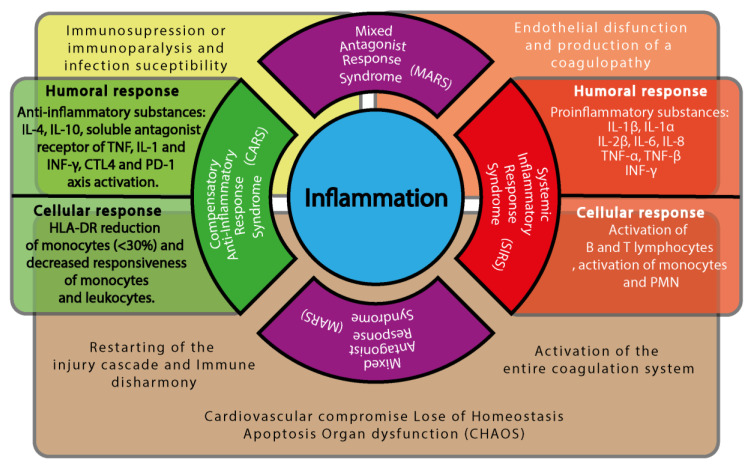
Immune cascade components and shock progression. This diagram illustrates the dynamic interplay between inflammatory and anti-inflammatory responses during shock, highlighting the transition from adaptive immune modulation to pathological dysregulation. At the center of the figure is a blue circle labeled “Inflammation”, representing the immunological core of the shock state. Surrounding it are four semicircular domains, each corresponding to a phase of the immune response. To the right, a red semicircle represents the Systemic Inflammatory Response Syndrome (SIRS), associated with the activation of the entire coagulation system. Adjacent to it, a red rectangular panel outlines the humoral response (e.g., pro-inflammatory mediators) and the cellular response (e.g., activation of neutrophils, monocytes, and endothelial cells). To the left, a green semicircle represents the Compensatory Anti-inflammatory Response Syndrome (CARS), linked to immune disharmony and the reactivation of injury cascades. A green rectangular panel describes the corresponding humoral response (e.g., anti-inflammatory cytokines) and cellular response (e.g., suppression of antigen presentation, T-cell deactivation). Above and below, two identical purple semicircles denote the Mixed Antagonist Response Syndrome (MARS), in which both pro- and anti-inflammatory responses coexist, leading to immune dysregulation. MARS is centrally associated with the development of CHAOS (Cardiovascular Compromise, Loss of Homeostasis, Apoptosis, Organ Dysfunction, and Immune Suppression). Behind the semicircular domains, a semi-transparent horizontal brown rectangle spans the figure, symbolizing systemic consequences that emerge as the immune syndromes overlap. Beneath CARS, the figure highlights immune disharmony and the reactivation of injury cascades. Beneath MARS, it shows the emergence of CHAOS. Beneath SIRS, it depicts coagulation system activation. Overlaying the upper portion of the immune domains are two semi-transparent bars that depict systemic clinical outcomes: A yellow bar above CARS and MARS indicates immune suppression or paralysis and increased susceptibility to infections. An orange bar above SIRS and MARS indicates endothelial dysfunction and the development of coagulopathy. Together, this figure emphasizes that these immune syndromes are not isolated stages but overlapping domains, each with its own systemic repercussions, which collectively shape the clinical trajectory of patients in shock.

**Figure 4 cells-14-00808-f004:**
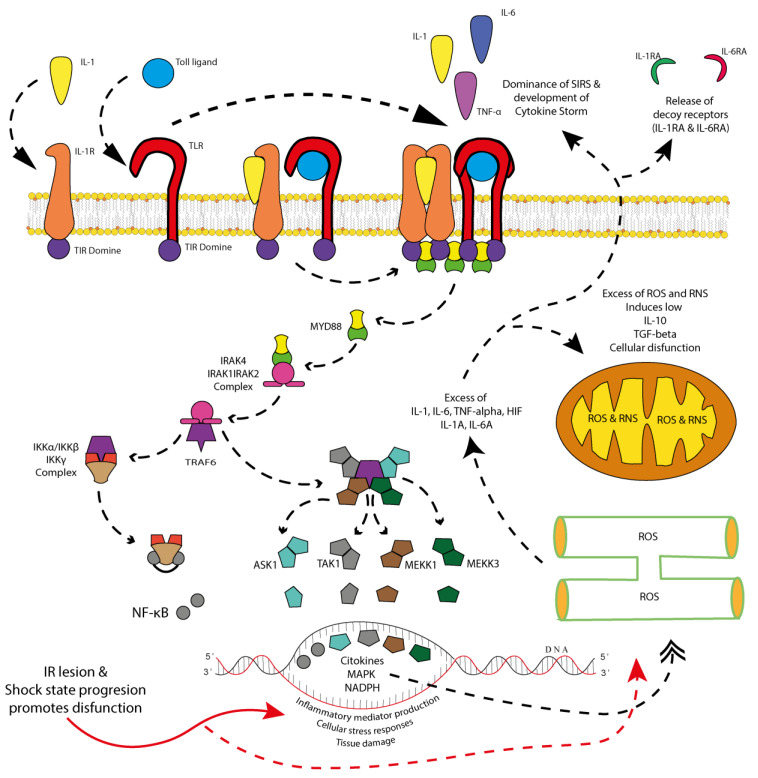
SIRS signaling: inflammatory pathways during shock and cytokine storm and the production of soluble cytokine receptor antagonists. Damaged tissue during the first hit releases pattern recognition receptors (PRRs), which initially activate Toll-like receptors (TLRs) on immune system cells (antigen-presenting cells, T cells, B cells, regulatory T cells, etc.). These, in turn, promote the release of pro-inflammatory cytokines, with the magnitude of this inflammatory response depending on both the extent of the first hit and the patient’s pre-existing comorbid state. Regardless of the scenario, the immune system initiates signaling pathways that involve the dimerization and activation of cytokine and Toll-like receptors (TLRs). Through their intracellular Toll/interleukin-1 receptor (TIR) domain, these receptors facilitate the formation of signaling platforms that enable the phosphorylation of MyD88. Once phosphorylated, MyD88 triggers the formation and activation of the IRAK4/IRAK1/IRAK2 complex, which subsequently interacts with TRAF6. TRAF6, in turn, activates two critical pathways: (1) the IKKα/IKKβ/IKKγ complex, leading to NF-κB phosphorylation and activation, and (2) the MAPK signaling pathway, involving ASK1, TAK1, MEKK1, and MEKK3. Both pathways synergistically enhance the cellular response to stress, promoting pro-inflammatory activity, oxidative stress (ROS and RNS) in the endoplasmic reticulum and mitochondria, and the release of inflammatory mediators. While this response is initially necessary for damage containment and tissue repair, its persistence can lead to an excessive pro-inflammatory state. Depending on the cell type undergoing this adaptive process (endothelial cells, epithelial cells, immune cells, or damaged cells) and the duration of shock progression, the inflammatory response may become overwhelming, leading to a dominant SIRS state. This results in an excessive release of pro-inflammatory cytokines, culminating in a cytokine storm, which, rather than being protective, exacerbates tissue damage and systemic inflammation. Graphical elements: Solid red arrows indicate the direct effect of ischemia/reperfusion injury and shock progression. Dashed red arrows represent the resulting intracellular oxidative damage. Black dashed arrows with triangular heads indicate signal transduction activation between receptors and intracellular mediators. Black dashed arrows with double heads represent cyclic organelle damage and feedback mechanisms. These arrows help visualize the direction and intensity of molecular events described above.

**Figure 5 cells-14-00808-f005:**
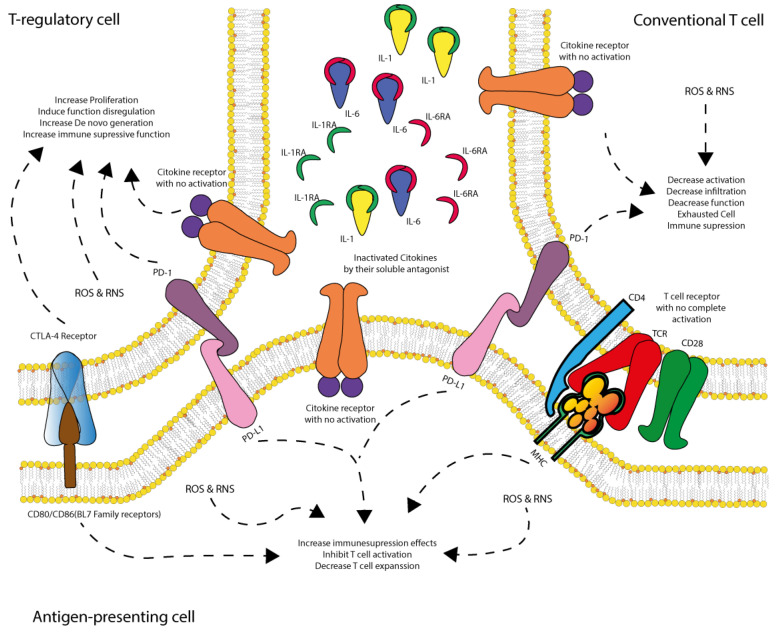
CARS regulation: role of dendritic cells in immune suppression and T cell inactivation. During SIRS activation following the first hit, immune regulatory mediators are progressively expressed to counteract excessive pro-inflammatory responses during shock. Dendritic cells (APCs) play a pivotal role in immune modulation; however, they are also susceptible to ischemia/reperfusion (IR) injury, oxidative stress, dysfunctional regulatory T-cells (Treg), and receptor reconfiguration in conventional T cells. As shock progresses, dendritic cells begin expressing PD-L1, which interacts with PD-1 receptors on dysfunctional Treg cells and conventional T cells. This interaction drives these cells into a non-responsive state or induces immune-suppressive activity, thereby directly and indirectly inhibiting T cell activation and expansion, ultimately leading to immune suppression and increased susceptibility to infections. Dysfunctional Treg cells express CTLA-4, PD-1, and inactive cytokine receptors due to cytokine sequestration by soluble antagonists (IL-1AR and IL-6AR). These cells exhibit oxidative stress (ROS and RNS) and abnormal proliferation, promoting the further generation of dysfunctional Treg cells and reinforcing immune suppression. Conventional T cells also exhibit inactive cytokine receptors due to cytokine sequestration, along with PD-1 expression and incomplete TCR activation due to the absence of CD28 co-stimulation (as its ligand, B7, is sequestered by CTLA-4 on Treg cells). These cells experience reduced infiltration, diminished function, and exhaustion, contributing to immune suppression. Dendritic cells (APCs), positioned between these T cell populations, have their B7 ligand sequestered by CTLA-4 on dysfunctional Treg cells, while also expressing PD-L1, which interacts with PD-1 on both T cell types. Additionally, they exhibit inactive cytokine receptors due to cytokine deprivation, further dampening T cell activation and limiting de novo T cell expansion. As a result, this immune dysregulation fosters a dominant CARS state, characterized by T cell dysfunction, immune suppression, and heightened vulnerability to infections.

**Table 1 cells-14-00808-t001:** Expanded categorization of shock types and clinical scenarios.

Shock Type	Subtype	Volume Mechanism	Tissue Injury	Clinical Scenario	Common Pathway
**Hypovolemic**	Hemorrhagic	Acute hemorrhage (critical)	No major soft tissue injury	Aortic dissection rupture	**Generalized tissue ischemia**
	T/hemorrhagic	Acute hemorrhage (critical)	With major soft tissue injury	Polytrauma	
	Pure hypovolemic	Critical reduction of plasma volume (fluid loss) without hemorrhage	No major soft tissue injury	Persistent fever, diarrhea, or vomiting	
	T/hypovolemic	Critical reduction of plasma volume (fluid loss) without hemorrhage	With major soft tissue injury	Large surface burns or deep skin lesions	
**Cardiogenic**	Ischemic	Decreased contractility/↓ cardiac output	Myocardial tissue injury	ST-elevation MI	
	Arrhythmic	Reduced ventricular filling or ejection due to abnormal rhythm	No direct structural injury	Sustained ventricular tachycardia	
	Valvular	Acute increase in preload or afterload due to valve dysfunction	Possible valve apparatus injury	Acute mitral regurgitation from chordae rupture	
	Myopathic	Progressive loss of myocardial pump function	Chronic myocardial injury	Decompensated dilated cardiomyopathy	
**Obstructive**	Pulmonary vascular	Obstruction of blood flow through pulmonary arteries/↓ left ventricular preload	No direct myocardial injury	Massive pulmonary embolism	
	Mechanical cardiac compression	Intrapericardial pressure limiting cardiac filling	Pericardial or pleural injury	Cardiac tamponade or tension pneumothorax	
	Outflow obstruction	Left ventricular ejection obstruction	Structural cardiac abnormality	Severe aortic stenosis	
**Distributive**	Septic	Vasodilation + capillary leak/relative hypovolemia	Inflammatory tissue injury	Sepsis with hypotension and elevated lactate	
	Anaphylactic	IgE-mediated vasodilation + increased permeability/plasma extravasation	Immune-mediated reaction	Bee sting or drug-induced anaphylaxis	
	Neurogenic	Loss of sympathetic tone/unopposed vagal tone and vasodilation	Spinal cord or CNS injury	Cervical spine trauma	
	Endocrinologic	Cortisol/thyroid hormone deficiency/vasodilation, impaired response to catecholamines	No structural tissue injury	Adrenal crisis or myxedema coma	
**Diss/Cyto**	Toxic-metabolic	Impaired cellular oxygen use despite adequate perfusion	Mitochondrial or enzymatic injury	Cyanide or carbon monoxide poisoning	

T/hemorrhagic = traumatic/hemorrhagic; T/hypovolemic = traumatic/hypovolemic.

**Table 2 cells-14-00808-t002:** Injury cascade progression and systemic immune response. This table outlines the sequential stages of immune response activation and progression during shock, highlighting the transition from initial inflammatory reactions (SIRS) to compensatory and maladaptive phases (CARS, MARS), culminating in immune dysregulation (CHAOS) and MODS due to I/R injury. SIRS: Systemic Inflammatory Response Syndrome; CARS: Compensatory Anti-inflammatory Response Syndrome; MARS: Mixed Antagonist Response Syndrome; MODS: Multiple Organ Dysfunction Syndrome; SOF: Single Organ Failure; MOF: Multiple Organ Failure; CHAOS: Cardiovascular Compromise—Loss of Homeostasis—Apoptosis—Organ Dysfunction (MODS, SOF, and MOF)—Immune Suppression.

Stage	Immune Syndrome	Trigger/Event	Dominant Immune Response	Systemic Consequences
**I**	SIRS	First hit (trauma, infection, ischemia)	Pro-inflammatory cytokines, immune cell activation	Initial containment, tissue repair initiation
**II**	CARS	Excessive inflammation or large injury	Anti-inflammatory mediators, immune suppression	Attempted immune balance, risk of suppression
**III-A**	MARS (SIRS over CARS)	Ongoing injury with dominant inflammation	Coexistence of pro- and anti-inflammatory states	Endothelial dysfunction, coagulopathy
**III-B**	MARS (CARS over SIRS)	Immune suppression becomes predominant	Immunoparalysis, decreased immune surveillance	Susceptibility to infection, reactivation of injury
**IV**	CHAOS	Failure of regulatory mechanisms	Total immune dysregulation, exhaustion	MODS, SOF, MOF, immune collapse

Note: Bold formatting in the first column is used to emphasize the numerical grade of each injury stage.

**Table 3 cells-14-00808-t003:** Theoretical implications of immune exhaustion pathways in shock and ischemia/reperfusion injury.

Immune Exhaustion Pathways						
**Pathway**	**Expression**	**Main Inducers**	Coupled Signaling Pathways	Cellular Effects	Effects of Overactivity/Inactivity	Ref.
**CTLA-4 PATHWAY** CTLA-4 COMPETES WITH CD28 FOR B7 LIGANDS (CD80/CD86) ON ANTIGEN-PRESENTING CELLS (APCS)	Induced after initial TCR activation but rapidly internalized in effector T cells. Constitutively expressed in Tregs.	TCR activation, IL-2, TGF-β, Treg differentiation.	Negatively regulates TCR signaling and costimulatory pathways via CD28-B7 interaction.	Prevents excessive T-cell activation, reduces inflammatory cytokine production, and maintains immune homeostasis.	Overactivity leads to excessive suppression of T-cell activation, reducing inflammatory cytokine production necessary for proper immune response and tissue repair. This can impair the clearance of pathogens and delay wound healing. Inactivity results in uncontrolled immune activation, increasing oxidative stress and tissue damage due to excessive pro-inflammatory cytokine release.	[248,249,250,251,252,253,254,255,256,257,258,259,260]
**PD-1 PATHWAY** PD-1 INTERACTS WITH ITS LIGANDS PD-L1 AND PD-L2, WHICH ARE EXPRESSED ON APCS AND SOME NON-IMMUNE CELLS	Induced in activated T cells, especially in response to chronic stimulation. Sustained expression in persistent infections.	Chronic TCR activation, IL-6, IL-10, TGF-β, hypoxia, IFN-γ.	Inhibits PI3K-Akt, Ras-MEK-ERK, and JAK-STAT signaling, reducing T-cell proliferation and cytokine production.	Suppresses T-cell proliferation, decreases cytokine production, and induces T-cell exhaustion in chronic infections and cancer.	Overactivity causes prolonged T-cell exhaustion, leading to reduced ability to control infections and impaired antioxidant defenses, increasing oxidative stress. This contributes to chronic inflammation and defective tissue regeneration. Inactivity results in excessive immune activation, enhancing ROS production, damaging tissues, and overwhelming reparative mechanisms.	

Note: This table reflects mechanistic associations from preclinical and clinical studies, highlighting pathways that may contribute to immune dysregulation during late-stage shock.

**Table 4 cells-14-00808-t004:** Clinical impact of oxidative stress in shock states.

Clinical Impactof Oxidative Stress in Shock States	
**Mechanism**	Clinical Impact
**Excessive ROS/RNS production**	Multi-organ dysfunction (MODS), endothelial damage, coagulopathy
**Uncontrolled NF-κB and inflammasome activation**	Cytokine storm in septic shock
**Loss of PD-1/CTLA-4 function**	Persistent immune activation, tissue destruction
**Treg/Th17 imbalance**	Chronic inflammation, increased susceptibility to secondary infections
**Reperfusion-induced ROS overload**	Worsening of ischemia/reperfusion injury, increased mortality

## Data Availability

No new data were created or analyzed in this study.

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
