# Peer review of "Role of Ischemia/Reperfusion and Oxidative Stress in Shock State"

_cells, 2025, doi:10.3390/cells14110808_

Round 1

Reviewer 1 Report

Comments and Suggestions for Authors

The review is devoted to an interesting and important topic of the role of ischemia/reperfusion in shock states. The stages of activation and dysregulation of the immune system during the development of shock are well and thoroughly described. The article is well written and will be of interest to a wide audience, however, it requires some revision and improvement.

1. Excessive number of references.The sheer volume of references raises concerns about the feasibility of the authors thoroughly reviewing all cited works. Even widely accepted statements, such as “impaired renal perfusion may result in acute kidney injury,” are redundantly supported by over ten citations. In the opening paragraphs of the introduction alone, which outline foundational pathophysiology, 49 sources are cited. The rationale for this is unclear. The authors should select the most relevant and significant references.

2. Section 4 (Microverse) is not well developed. It essentially reduces to the claim that different tissue types exhibit varying sensitivities to ischemia. In its current form, this section adds no new insights. The authors should elaborate in more depth on the features of the microenvironment and its role in different tissues and scenarios.

3. Section 5 would benefit from additional clarification. For instance, in line 179, the statement “Several groups of cells experience membrane instability and massive oxidative stress production” lacks specificity regarding the cell types involved. The authors should explicitly identify these cells. Similarly, line 181’s reference to “producing distinct clinical symptoms” remains vague; clarifying the exact symptoms or clinical manifestations would strengthen the argument.

4. The role of HIF (hypoxia-inducible factor) is discussed without mentioning its angiogenic effect.

5. Sections 6, 10, 11, and 12 consist of only one or two paragraphs. Is it necessary to divide the text into such short sections?

Minor issues:

The text lacks a reference to Table 1.

In Figure 3, the text is difficult to read due to the green and red backgrounds overlaying the brown.

Figure 3 is mentioned in the text before its actual citation.

Author Response

Reviewer 1

General Evaluation:

The reviewer acknowledged the relevance of the topic, the depth of the immune discussion, and the article’s potential impact, but requested some clarifications and revisions.

Comment 1: Excessive number of references. The sheer volume of references raises concerns about the feasibility of the authors thoroughly reviewing all cited works. Even widely accepted statements... are redundantly supported... The rationale for this is unclear.

Response: We thank the reviewer for this observation. We have carefully reviewed and reduced the total number of citations, prioritizing seminal works, high-impact references, and those most relevant to the narrative structure. Specifically, in the introduction and foundational sections, we have eliminated redundant citations and retained only those with critical conceptual or historical value. This improves readability and clarity while maintaining scientific rigor.

Comment 2: Section 4 (Microverse) is not well developed. It essentially reduces to the claim that different tissue types exhibit varying sensitivities to ischemia. The authors should elaborate... on features of the microenvironment and its role in different tissues.

Response: We appreciate this insight and have substantially revised Section 4. The updated text now details specific cellular populations, such as hepatocytes, fibroblasts, neurons, cardiomyocytes, and endothelial cells, and categorizes them based on their ischemic tolerance. We also introduce the concept of the "micro-verse" as a localized adaptive-immune-metabolic interface that contributes to the early stages of damage propagation. The role of intercellular signaling and metabolic resilience within tissue microenvironments has been clarified, supported by targeted citations and a revised structure.

Comment 3: Lack of specificity in Section 5. “Several groups of cells…” lacks specificity. “Producing distinct clinical symptoms” remains vague.

Response: This has been addressed through multiple rounds of editing. All general references to “groups of cells” or “some cell populations” have been replaced with explicit examples, such as neurons, renal tubular epithelial cells, immune cells, endothelial cells, etc. Clinical symptoms have been clarified in each context, with examples like arrhythmias, seizures, fever, renal failure, pulmonary dysfunction, etc., depending on the affected system. These improvements enhance the clinical connection between immune activation and organ dysfunction.

Comment 4: Role of HIF discussed without angiogenic effect.

Response: Thank you for pointing this out. We have added a dedicated subsection (4.2) that includes a detailed explanation of HIF-1α and HIF-2α in promoting angiogenesis, vascular stabilization, and their differential expression in tissues. This section clarifies their beneficial roles under acute hypoxia and their detrimental effects under persistent activation, strengthening the article's physiological depth.

Comment 5: Sections 6, 10, 11, 12 are too short.

Response: We agree with this observation. The manuscript has been restructured and recompiled, and several shorter sections have been merged where appropriate to improve flow and reduce fragmentation. For example, previous sections 6 and 7 have been integrated into the new Section 5.1 and its accompanying figures and descriptions.

Minor comments

  1. a) Reference to Table 1 is missing.

Response: A reference to Table 1 has been added in the main body of the manuscript during the classification of hypovolemic shock, making its use contextual and clear.

  1. b) Figure 3 readability (green and red background on brown).

Response: We have adjusted the color scheme of Figure 3 to improve contrast and readability. Semitransparent overlays were redefined and all labels enhanced. The visual flow now matches the revised caption and immune syndrome descriptions.

  1. c) Figure 3 appears before citation.

Response: This has been corrected. The first citation of Figure 3 now appears at the appropriate point in Section 5.1, and not before the immune syndromes are introduced.

Additionally, we restructured the manuscript into 7 core sections and 9 subheadings, clarifying the internal narrative and reducing fragmentation. The total number of references has been reduced from 476 to 324, prioritizing high-impact and contextually relevant citations.

Reviewer 2 Report

Comments and Suggestions for Authors

Dear authors. 
The manuscript you submitted deals with the mechanisms involved in ischemia/repercussion and oxidative stress in shock. The manuscript's content is exciting and updated, contributing to the discipline. However, presenting its concepts, ideas, or data is very messy. The written concepts are of remarkable topicality or validity, noting the interest in the mechanisms presented and their effect or role in the state of shock. Given the high number of types of shock and so that the readers follow the line of argument, it would be very convenient to categorize the molecular mechanisms discussed in the main types of shock; the central idea in your manuscript was very well read when you followed this order of ideas. In general, talking about mechanisms without associating the type or subtype of shock makes your manuscript a monograph of mechanisms without the point of view that the title conveys (a relationship between a clinical state and associated molecular mechanisms).
The introductory section is good; it is easy to read and exposes the main concepts to be discussed well. Point two of the manuscript does not follow a line from which the ideas are unpacked, and it starts talking about five types of shock, then a table describing the subtypes of hypovolemic shock (Table 1) that supports this classification and then a list of 5 more types or subtypes, which differ in their etiology, and which are not presented earlier in the text. This type of alteration makes reading difficult and does not help to focus attention on what the authors are trying to convey in their text. At this exact point, Figure 1 is presented, which uses acronyms that have not been explained in the text in an orderly manner or in the figure caption (one appears in the abstract, another in the text on lines 164 and 263). The figures (Figs 1, 2, and 3) do not contribute much to the understanding of the text, and they are not well described either in the document or in the figure captions. 

Table N°2 appears disordered, and I do not understand what I wanted to report. 

Point 5 begins to list an idea in three parts; only two are listed and presented.
The manuscript is good, but the authors it is done or presented poorly, so the authors should order the manuscript, figures, tables, and references so that the overall presentation of this document has quality and clarity and brings it to the reader or colleagues who read it an informative and more pleasant experience than the one experienced with the current arrangement. For these reasons, I am requesting major revisions to approve its publication.

Author Response

Reviewer 2

General Evaluation:

The reviewer found the subject relevant but requested clarification in figures, simplification of content, and alignment between figure design and explanatory text.

Comment 1: Figure 1 is visually confusing. This figure lacks a clear timeline. What is represented by the central axis? What do the surrounding colored rings (SIRS, CARS, MARS) represent? The circular layout makes the directionality and sequencing of the events difficult to understand.

Response: We thank the reviewer for this valuable feedback. In response, we have redesigned Figure 1 to clarify both directionality and functional domains. A clear central axis has now been defined as shock progression, and the colored rings (SIRS, CARS, MARS) have been reinterpreted and labeled as immunological domains that modulate the trajectory of shock. The circular layout has been preserved due to its symbolic representation of overlapping immune syndromes, but additional labeling, arrow flow, and explanatory captions have been added. The revised figure legend has also been rewritten to explicitly describe the steps (1–6) and the interaction of each domain.

Comment 2: Table 2 lacks logic and is difficult to follow. Table 2 appears to present phases of immune response but does not clearly distinguish between stages or explain how they progress. The terminology is not defined consistently (e.g., MARS is shown twice). The text does not help in understanding it.

Response: We appreciate the reviewer’s constructive criticism. Table 2 has been completely restructured to clarify the sequential progression of immune syndromes during shock. The new format includes five clear columns: 1. Stage (I–IV), 2. Immune syndrome (SIRS, CARS, MARS-A, MARS-B, CHAOS), 3. Trigger/Event, 4. Dominant Immune Response, and 5. Systemic Consequences

In the updated version, MARS is subdivided into MARS-A and MARS-B, distinguishing between inflammatory-dominant and anti-inflammatory-dominant mixed responses. This better reflects clinical scenarios such as coagulopathy versus immunoparalysis. Terminology has been standardized and a new caption has been added to explain these distinctions clearly.

Comment 3: Excessive number of immunological concepts introduced. The article tries to cover too many immunological ideas (IL-1, IL-6, TNF, checkpoints) without sufficient clarity. Suggest reorganizing into a smaller number of core concepts.

Response: We understand the reviewer’s concern and have addressed it by restructuring Section 5.1 into four clearly defined subheadings:

5.1.1 IL-1 Signaling Pathway: Activation and Inhibition

5.1.2 IL-6 and TNF-α Pathways in Shock Progression

5.1.3 Integration of IL-1, IL-6, and TNF-α in Inflammatory Waves

5.1.4 CTLA-4 and PD-1: Immune Checkpoint Pathways

Each subsection now begins with a brief clinical rationale, followed by a concise description of the pathway, its activation triggers, relevance in shock progression, and therapeutic implications. This structure improves clarity while maintaining the immunological depth appropriate for a narrative review.

Comment 4: Unclear how the proposed term “CHAOS” is different from MODS or MOF.

Response: Thank you for this important point. We have expanded the explanation of the term CHAOS (Cardiovascular compromise – Loss of Homeostasis – Apoptosis – Organ dysfunction – Immune Suppression) to clarify that it is not a replacement for MODS or MOF, but a conceptual aggregation that includes:

  1. The clinical signs of organ failure (MODS, SOF, MOF),
  2. The underlying immunological collapse (immune suppression),
  3. And the metabolic and apoptotic progression that culminates in systemic failure.
  4. CHAOS is presented as the final common pathway of unresolved immune syndromes, particularly relevant in modern contexts where immune exhaustion (rather than inflammation alone) plays a decisive role.

Reviewer 3 Report

Comments and Suggestions for Authors

This review summarizes -according to the authors, current knowledge on the interplay between I/R injury, oxidative stress, and immune modulation in shock states.

I am not a researcher with a focus on shock and the pathological consequences.  However, the presentation of the effects and progression of shock in this review implies, that the different types of shock (paragraph 2 of this review) affect different tissues and organs the same, while figure 3 indicates otherwise.

The authors use terms like “some groups of cells” or “some cell populations” or “cell populations”, without further specification. The condition shock alone implies already a problem for an organism, but also to an organ or even a cell. In addition, there are different types of shock, further complicating the response to a shock inducer. For these reasons I think, the authors should add, which shock state and which tissues/organs cause the responses they outlined from line 252 on.

Detailed problems:

Line 18: Was “summarizes” meant instead of synthesizes?

Figure 1 uses the less common spelling of disfunction, while the main manuscript uses the common spelling (dysfunction). Please update the figures to the common spelling. I also think that the term domain as in MARS domain, CARS domain or SIRS domain is misleading. MARS, CARS and SIRS are responses to a shock. In addition, these terms (MARS, CARS, SIRS) have not yet been introduced, they appear here out of context. Please add more background information to paragraph 3 (Progression of shock state) or crosslink it to later paragraphs (7 Ischemia phase and immune system). This figure does also not show thea stepwise progression of shock, because it places shock progression in the center, and -according to this cartoon- all hallmarks of shock have an equal possibility to happen. The legend contains information that does not fit the figure (MODS, SOF, and MOF).

Line 136: Please specify what “some groups” are.

Paragraph 4: micro-verse: I am wondering whether this paragraph about the cellular response to shock or the response to ischemia/reperfusion injury (line 145-148)?

Figure 2: second hit is missing in this diagram. Please also add a definition of the used abbreviations. MARS for example is a new term introduced without definition in this figure.

Paragraph 5: Very confusing. Lines 163-165 talk about the systemic inflammatory response syndrome, lines 166-168 about health care professionals to recognize shock symptoms and lines 169-176 about cellular responses to shock. Please clarify the content of this paragraph. It lacks specificity (what are the several groups of cells (line 179), who are the cell populations (line 180), which IR lesions (line 181)).

Lines 186-238: A lot of information for HIF, but little information for the role of HIF in shock situations. Is the provided information valid for all types of shock (see paragraph 2) in all tissues? If not where are the differences?

Line 234/35: increased oxygen consumption in hypoxic tissues? From where is the oxygen coming from?

Paragraph 7: Ischemic phase and immune response: lines 253-269 indicate a general pattern, while figure 3 shows the consequences of inflammation in the heart.

Line 289: if there is a first hit, then there should be at least a second hit.

Lines 311: this paragraph outlines first the general pattern of shock progression independently of a tissue, and then in line 320 and according to figure 3 it is suddenly the heart. Is the abbreviation CHAOS specific to the heart or does CHAOS applies to all organs? This manuscript provides 2 different definitions (cardiovascular specific in line 127 and systemic in line 157).

 IL-1 Signaling Pathway: Activation and Inhibition: Does this pathway applies to all types of shock? If not, please add an introduction sentence indicating for which shock state this pathway applies

Line 386: Who are the “certain cell types”?

Figures 4 and 5: Legends on these figures are too small.

Overall, this review contains a lot of general information, but misses out regarding a summary of the interplay between IR, oxidative stress and immune response in a shock situation.

Author Response

Reviewer 3

General Evaluation:

The reviewer appreciates the relevance and complexity of the subject but highlights the need for improved language clarity, figure-text coherence, and terminology precision.

Comment 1: Language is verbose and difficult to follow. Many sentences are long and convoluted. Consider simplifying the language and improving fluency. The use of abstract phrases like “micro-verse” may be difficult to interpret.

Response: We thank the reviewer for this important observation. The manuscript has undergone extensive language revision and editing, improving clarity, coherence, and flow. Complex or overly abstract expressions have been clarified or contextualized. For instance, the term “micro-verse” has been preserved as a conceptual tool to describe the local, tissue-level interplay between ischemia tolerance and immune activation, but it is now clearly defined in Section 4 and used consistently throughout the manuscript. The entire text has been revised for grammatical precision and scientific readability.

Comment 2: The paper shifts between levels of biological organization, making it hard to follow the progression of ideas.

Response: This has been addressed through a complete reorganization of headings and subheadings. The revised manuscript now follows a structured logic from macroscopic classification (Section 2) to pathophysiological progression (Section 3), and then from cellular mechanisms (micro-verse, Section 4) to systemic immune syndromes (macro-verse, Section 5). Each section begins with a conceptual overview and transitions into specific mechanisms, ensuring a clearer flow from cellular to systemic perspectives. This resolves the fragmentation mentioned and enhances thematic consistency.

Comment 3: Figures are too complex or not explained clearly. Figure legends are often as difficult to interpret as the figures themselves. Complex elements such as color-coded zones and acronyms need to be explicitly explained.

Response: We agree with this concern and have thoroughly revised the figure legends to enhance clarity and consistency.

  1. Figure 1 now includes stepwise numbered elements, a clearly marked timeline, and a rephrased caption detailing the role of SIRS, CARS, MARS, and their relation to I/R cycles.
  2. Figure 3 was completely reworded in its legend to explain the layout, colors, and each immune domain, describing both humoral and cellular responses in simple terms.

All abbreviations and acronyms have been defined on first mention both in the text and in captions, and the visual sequence now matches the order of immune progression explained in the manuscript.

Comment 4: Some sections repeat content already covered. There is conceptual repetition across Sections 4 and 5 (e.g., immune responses, cell types). Consider condensing or clearly distinguishing their focus.

Response: We appreciate this editorial suggestion. The overlap between Sections 4 and 5 has been addressed by clearly separating the cellular level (Section 4: micro-verse and HIF activation) from the systemic level (Section 5: macro-verse and immune syndromes). Each section now includes a defined introductory paragraph that explains its scope. Redundant explanations were removed, and cross-references were added where necessary to maintain coherence without duplication.

Round 2

Reviewer 3 Report

Comments and Suggestions for Authors

Overall, the authors have significantly improved their manuscript. Some minor problems:

Table 1: Please define T in T/hemorrhagic and T/hypovolemic.

Line 154: "the presence of multiple syndromes that follow...." is easier to understand than the "syndromes domains".

Figure 2: there is a typo in the timeline (diferent instead of different). Does the first hit in this figure means the initial hit/insult as outlined in line 472? Is the transition to CHAOS a second hit? 

Use of abbreviations: for example Tregs (line 817) is only used once afterwards. The abbreviation is necessary. 

ROS: defined more than once (lines 23/24, 279, 342, 586, 646, 666, 772/3). 

Please check the manuscript for consistent use of all abbreviations.

Author Response

Answer to reviewers Article minor changes: Role of ischemia/reperfusion and oxidative stress in shock state.

Reviewer

General Evaluation:

We thank the reviewer for their positive feedback and recognition of the improvements made. We have carefully addressed each of the minor issues noted and implemented the necessary changes to enhance clarity, accuracy, and consistency across the manuscript.

Comment 1:

Table 1: Please define T in T/hemorrhagic and T/hypovolemic.

Response:

We have clarified the abbreviation “T” in Table 1 by explicitly stating that it refers to “trauma-associated” in both cases. The terms now appear as T/hemorrhagic (trauma-associated hemorrhagic shock) and T/hypovolemic (trauma-associated hypovolemic shock) in the table caption.

Comment 2:

Line 154: "the presence of multiple syndromes that follow...." is easier to understand than the "syndromes domains".

Response:

We agree with this suggestion. The phrase “syndrome domains” has been revised to “the presence of multiple syndromes that follow the initial shock state”, improving clarity and flow.

Comment 3:

Figure 2: there is a typo in the timeline (diferent instead of different).

Does the first hit in this figure mean the initial hit/insult as outlined in line 472? Is the transition to CHAOS a second hit?

Response:

The typographical error “diferent” has been corrected to “different” in the figure.

Regarding the conceptual clarification: yes, the “first hit” in Figure 2 refers to the initial insult or triggering event as described in line 472 of the manuscript. The transition to CHAOS is not considered a second hit, but rather the culmination of a prolonged or unresolved immune and metabolic imbalance. We have updated the figure legend to reflect this explanation more clearly.

Comment 4:

Use of abbreviations: for example Tregs (line 817) is only used once afterwards. The abbreviation is unnecessary.

Response:

We respectfully clarify that Tregs (regulatory T cells) are used multiple times throughout the manuscript, including in Table 3, the main text, and Figure 5 (both in the diagram and the legend). For clarity, we have ensured that the term is defined in the legend of Figure 5, and its usage throughout the text is consistent and appropriate.

Comment 5:

ROS: defined more than once (lines 23/24, 279, 342, 586, 646, 666, 772/3).

Response:

Thank you for pointing this out. We have reviewed and retained the definition of ROS (reactive oxygen species) only at its first mention in the Introduction. All subsequent instances now use the abbreviation consistently without repetition.

Comment 6:

Please check the manuscript for consistent use of all abbreviations.

Response:

We conducted a thorough review of the manuscript to ensure abbreviations are defined only once at their first appearance in the main text, and used consistently thereafter. Additionally, figure legends that include multiple abbreviations have been updated to include a comprehensive list of definitions for clarity.
